# Functionality of Cordia and Ziziphus Gums with Respect to the Dough Properties and Baking Performance of Stored Pan Bread and Sponge Cakes

**DOI:** 10.3390/foods11030460

**Published:** 2022-02-03

**Authors:** Mohamed Saleh Alamri, Abdellatif A. Mohamed, Shahzad Hussain, Mohamed A. Ibraheem, Akram A. Abdo Qasem, Ghalia Shamlan, Mohammed Jamal Hakeem, Ibrahim A. Ababtain

**Affiliations:** Department of Food Science and Nutrition, King Saud University, Riyadh 1145, Saudi Arabia; msalamri@ksu.edu.sa (M.S.A.); shhussain@ksu.edu.sa (S.H.); mfadol@ksu.edu.sa (M.A.I.); aqasem@ksu.edu.sa (A.A.A.Q.); shamlana@ksu.edu.sa (G.S.); mhakeem@ksu.edu.sa (M.J.H.); ababtain.ibr@gmail.com (I.A.A.)

**Keywords:** natural gums, baking, dough, pasting, cake, bread

## Abstract

The functionality of hydrocolloids of different origins, gum Cordia (GC), and gum ziziphus (GZ) on pan bread and sponge cake quality and their potential use in retarding the staling process have been studied. The effects of the gums were determined by assessing the pasting qualities of wheat flour slurry, dough properties, and the finished product. After 24 and 96 h of storage, investigations were conducted on the finished product. Micro-doughLab was used to assess dough mixing qualities, and a texture profile analysis (TPA) test was used to assess the texture. A hedonic sensory test of texture, scent, taste, color, and general approval was also conducted. The type of gum used had a significant impact on the physical properties of the bread and cake and their evolution through time. Reduced amylose retrogradation was demonstrated by the lower peak viscosity and substantially lower setback of wheat flour gels, which corresponded to lower gel hardness. Gums were superior at increasing the bread loaf volume, especially GZ, although gums had the opposite effect on cake volume. After both storage periods, the hardness of the bread and cake was much lower than that of the control. Except when 2% GC was used, adding GC and GZ gums to bread and cake invariably increased the overall acceptability of the product. In terms of shelf-life, GZ was able to retain all texture parameters, volume, and general acceptability close to the control after storage.

## 1. Introduction

The use of additives in the baking industry has become prevalent in recent years. Their use intends to optimize dough handling properties, improve fresh bread quality, and lengthen the shelf life of stored bread. With this objective in mind, a wide range of additives with various chemical structures are used. A group of additives that is commonly used in the food industry but not in baking is hydrocolloids. When modified cellulose (carboxymethylcellulose and cellulose derivatives) were coupled with monoglycerides, enzymes, and emulsifiers, dough plasticity was reduced, according to [1]. Hydrocolloids are utilized to improve dough performance, bread characteristics, and sensory quality. They have also been shown to prevent the crumb texture from changing during storage (antistaling effect). In addition, they can aid in the preservation of baked items’ structure, resulting in acceptable products [2]. Gums are a class of hydrocolloids compounds that stabilize emulsions, suspensions, and foams, which can affect the rheology and texture of aqueous systems [3]. Gums have been shown to be effective antistaling agents, especially when utilized in new technologies such as BOT (baking off technologies). By applying these improvers, they can compensate for the damage caused by dough freezing [4]. After the addition of 0.2% xanthan gum, the water absorption of the dough increased by 1.6%, while the mixing tolerance index and degree of softening decreased by 40 Brabender Units (BU). Gum had no effect on the gelatinization temperature or the pasting peak of the starch when applied at a 0.2% level or less [5]. Hydrocolloids are expected to increase water retention and loaf volume while decreasing stiffness and starch retrogradation when used in small amounts (1%) [6]. The very hydrophilic characteristic of hydrocolloids also aids in the prevention of ice crystal development and water migration during frozen storage, which improves the freeze/thaw stability of the dough or finished product [7]. Although varied gains were noted for different hydrocolloids, hydroxypropylmethylcellulose (HPMC) was the hydrocolloid that improved all of the measures examined, including the specific volume index and crumb hardness, as well as reducing moisture loss during bread storage and the crumb dehydration rate [8]. The ability of HPMC ability to act as a bread improver is attributed to its hydrophilic structure, which allows it to interact with water, as well as to increase the interface activity between water and the non-aqueous phases of the bread dough, which favors the formation of emulsions and strong uniform films [9]. Some hydrocolloids, such as locus bean, are said to have a softening effect on baked goods due to their high water retention capacity, or the prevention of gluten–starch interactions in the case of xanthan gum and alginate [10]. Although the vast use of hydrocolloids in breadmaking is obvious from the above, the qualities of hydrocolloids vary greatly depending on their origin and chemical structure [11]. Hydrocolloids of various chemical structures (xanthan sodium alginate and a k-carrageenan) altered the rheological qualities of wheat flour doughs to varying degrees. At the dough level, xanthan gum had the greatest impacts since this hydrocolloid was found to reinforce the wheat flour dough network. In addition, fermentation experiments demonstrate that xanthan has adequate qualities to be utilized as an additive when extensive fermentation processes are desired [12,13].

In the presence of carboxymethylcellulose (CMC), gellan, xanthan, guar gum, and sodium alginate, a viscosity rise was observed before reaching the typical starch pasting temperature. Waxy maize, waxy rice, tapioca, regular rice, potato, and wheat starches all produced different results in the presence of gums. The viscosity rise that occurred before starch pasting appeared to be caused by interactions between specific leached-molecules, particularly amylose, and certain gums [14]. Therefore, the pasting characteristics of the starch are dependent on the type and level of hydrocolloids, electrical charges available, starch source, and the method of measurement used. Consumer acceptability of cake depends on its volume and is controlled by the batter viscosity. If the viscosity of the batter is too low, air bubbles will escape and the cake will collapse in the oven. Highly viscous batters, on the other hand, keep air bubbles within, but prevent expansion of the bubbles [15]. Increases in cake volume and batter viscosity were confirmed when the concentration of gums (β-glucan) was increased to a given level. When the β-glucan gum concentration was increased further, the viscosity increased but the volume decreased. This could be attributed to the inability of air bubbles to expand with greater viscosities [16,17,18]. In this study, the quality parameters of gum Cordia (GC)– and gum Ziziphus (GZ)–flour blends were investigated. The impact of these gums on the rheological and physical properties of wheat flour dough was evaluated. The major purpose of this study was to compare and contrast the characteristics of these mixes in terms of bread and cake during storage.

Gum Cordia has a pseudoplastic tendency and a high viscosity, making it ideal for use in the food and pharmaceutical industries as a powerful emulsifier, thickener, and stabilizer [19]. Previous research has found that gum Cordia influences the physicochemical properties of apple jelly and significantly enhances its phenolic component concentration [20]. Gel samples containing 5% gum Cordia had similar rheological properties to the control but had the highest apparent viscosity, loss moduli (G″), storage moduli (G′), and complex viscosity [21]. Cordia fruit has an edible coating supplemented with CaCl_2_ or ascorbic acid, which has a significant positive effect on weight loss, total phenolic compounds, as well as inhibition of polyphenol oxidase activity [22]. Previously published data reported the successful utilization of gum Cordia as a coating on nuts to delay oxidative rancidity and moisture loss [19,23]. *Zizyphus Spina-Christi* is a tree species that belongs to the Rhamnaceae family of plants. It grows in a wide area of Africa, from Mauritania to the Red Sea. Literature states that the ethanol extract from *Ziziphus* fruits had rheological properties that were similar to xanthan gum and better than those of guar gum [24]. *Ziziphus* mucilage should be extracted with 1:7 water at 60 °C, and 1:3 ethanol should be used to precipitate it. When it comes to water holding, oil absorption, and emulsifying, the water holding capacity was 73.35 g water/g dry basis, 4.97 g oil/g dry sample, and 52.22% of the dry sample, respectively [25,26]. These data show that gum ziziphus can absorb more oil than guar gum or xanthan gum, but with a lower ability to make emulsions. The shape, size, color, and weight of the fruits of most *Ziziphus* varieties are different [27]. Caffeic acid and p-coumaric acid are two of the five phenolic compounds found in the extract of ziziphus, in addition to three other compounds [28]. Many researchers are focused on finding new sources and new applications of natural gums. The purpose of this research was to isolate polysaccharides from *Ziziphus* and *Cordia myxa* fruits. The isolated gums will be added to panned-bread and cake formulations. Therefore, the study will explore how the rheological and physical qualities of wheat flour dough and the physicochemical qualities of panned-bread and cake will be affected by the addition of these gums.

## 2. Materials and Methods

### 2.1. Isolation of gums

Gum Cordia (GC) and gum Ziziphus (GC) were extracted according to the methods described in another study carried out by the same research group [29]. To prevent enzymatic browning, cardia or ziziphus fruits were destoned, carefully washed, and steamed for 3 min. Pulp was made by blending the fruits at high speed for 1 min in 25 °C distilled water at 1:3 ratios in an auxiliary kitchen mixer (BioloMix, Whirlpool corporation, Benton Harbor, MI, USA), filtered through muslin cloth, and centrifuged at 10000× *g* for 30 min (Fisherbrand^TM^ Refrigerated Centrifuge GT2). The supernatant was neutralized, freeze-dried, and sieved through a 60 mesh sieve and stored at 4.0 °C. According to the weight of the fruits, the gum yield was 1.8% and 11.5% for GC and GZ, respectively.

### 2.2. Preparation of wheat flour and gum blends

The freeze-dried gum Cordia and gum Ziziphus powders were mixed with wheat flour at a rate of 1 and 2% of the flour weight. The use of 1 and 2% of the gums was selected based on past research experience of other researchers. A number of tests were performed on flour mixtures prior to baking. Tests performed on the flour-gum blends included RVA, gel texture, dough mechanical rheology, bread and cake firmness, and bread and cake sensory evaluation.

### 2.3. Rapid Visco analyzer measurements (RVA)

The Rapid Visco Analyzer was used to test the pasting properties of the flour mixtures (Newport Scientific, Sydney, Australia). The sample (3.5 g) was placed into RVA canisters, and distilled water was added to achieve a total weight of 28.5 g. The slurry was maintained at 50 °C for 50 sec, then heated to 95 °C at a rate of 12.16 °C/min and held for 5 min, and then cooled to 50 °C in 2 min and held for 2 min. The RVA profile provided the flowing pasting parameters: peak viscosity, final viscosity, breakdown, setback, and pasting temperature. Thermocline window software, provided by the manufacturer, was used to analyze the data [30].

### 2.4. Gel Texture

The texture of flour gels obtained from RVA canisters was determined by transferring the gel to a 25 mL beaker and keeping it at room temperature (25 °C) overnight. A Brookfield CT3 Texture Analyzer (Brookfield Engineering Laboratories, Inc. Middleboro, MA, USA) fitted with a 12.7-mm-wide and 35-mm-long cylindrical probe that was used to compress the gels in two penetration cycles at a speed of 0.5 mm/s for a distance of 10 mm. In the two penetration cycles, the gels were compressed at a speed of 0.5 mm/s to a depth of 10 mm using a TA-TXT texture analyzer cylinder (Vienna Court, Lammas Road, Godalming, Surrey, UK). Hardness, cohesiveness, and gumminess were determined from the measured gel properties [31].

### 2.5. Dough Mixing Properties Assessed using DoughLab

The Micro-doughLab (Perten Instruments, Sidney, Australia) was used to determine the optimum water absorption capacity to reach a peak of 500 FU using a 4.00 ± 0.01 g sample at a 14% moisture basis after moisture correction. For 20 min, samples were mixed at a speed of 63 rpm and at a temperature of 30 °C. For each sample, measurements were taken at least three times. The dough development time (min), stability (min), softening (FU), mixing tolerance index (MTI) (FU), and quality number were calculated using the mixing curve.

### 2.6. Bread Baking Procedure

Baking performance of the control flour and the blends was determined by method No. 10-09 [32]. Dry ingredients were added based on the flour weight; flour control (100% wheat flour) or with gum powders (1% or 2% GC or GZ; 100 g), 3 g instant dry yeast, 6 g sugar, 0.02 g improver (α-amylase and vitamin c), 4 g fat-free instant milk powder, 1.5 g salt, and 5 g shortening. The dough was prepared using the appropriate amount of water absorption for each blend and mixed accordingly. The straight dough method was used to prepare the panned bread. The dough was proofed for 60 min before punching and then for another 30 min after punching. After the second proofing, the dough was sheeted, molded, panned, and proofed for another 30 min before baking. All bread loaves were baked in a hearth-type oven for 20 min at 220 °C (National MFG. Co., Lincoln, NE 68508, USA). Baked loaves were allowed to cool and were then sliced and stored for further testing.

### 2.7. Cake Baking Procedure

The cake was baked according to the AACC method No. 10-90 [32]. Flour control (100% wheat flour) or with gum powders (1% or 2% GC or GZ; 100 g), 100 g sugar, 20 g shortening, 240 g fresh eggs, and 2 g baking powder were used in the cake. The sponge cake was baked in a hearth-type oven (National MFG. Co., Lincoln, NE 68508) at 190 °C after all components were mixed according to the method instructions. The cooled cake was stored until further use.

### 2.8. Bread and Cake Firmness after Storage

Bread samples were stored at room temperature, and firmness was measured after 24 and 96 h using two central loaf slices (25 mm thick) according to AACC method No. 74-09 [32]. A cylindrical probe (20 mm) mounted on a TA-TXT texture analyzer (Stable Micro Systems, Surrey, UK) equipped with a 50 kg compression cell was used. A 25% compression of a 25-mm-thick bread sample and a 6.25 mm compression depth were applied; at that point, the compression force value (CFV) was taken as the firmness, and the data were processed using Exponent software provided by the Stable microsystem. With the use of the same probe, a 60 mm cake slice, from the middle, was compressed at a 25% strain. The probe was kept at this distance for 60 sec before being removed from the sample and returned to its initial position, allowing the percentage of springiness to be calculated. For both product tests, the pretest and test speeds were set at 1.0 mm/sec.

### 2.9. Crumb Color of Bread and Cakes

Color parameters such as L*(Lightness), a* (Redness), and b* (Yellowness) of bread and cake crumb samples were determined using a Chroma meter Minolta color grader with a D65 light source (Konica Minolta CR-40) [33].

### 2.10. Sensory Evaluation

Sensory evaluation of bread samples was performed by a trained panel of judges using a 9-point hedonic scale. The panelists (12 members) were trained by providing them control bread and asking them to distinguish between control bread samples manufactured with different formulations. They were members of the department’s staff, graduating students, and laboratory specialists. The sensory panel was made up of panelists who were successful in detecting which bread or cake samples were similar and which were different. Because bread is commonly consumed in this society, all of the panelists are familiar with it. The selected panelists (seven members) were asked to compare and contrast the volume, texture, crust color, aroma, taste, crumb color, and overall acceptability of the samples and the control. Each of these sensory criteria was described to the panelists and they were given several trials to master them. The same process was applied for cake sensory evaluation, but the panelists (six members) were trained and asked to compare and contrast the texture, porosity, crumb color, taste, scent, and general acceptability of the cake samples relative to the control.

### 2.11. Statistical Analysis

The data were analyzed using ANOVA after the measurements were done in triplicate. The effects of GC and GZ on bread and cake were investigated using a factorial design. With the use of PASW^®^ Statistics 18 software, Duncan’s multiple range test was used to compare means at *p* ≤ 0.05. (SPSS Inc., Hong Kong, China).

## 3. Results and Discussion

### 3.1. Pasting Properties of Flour Blends (RVA)

The pasting properties and profiles of the flour composites are presented in Table 1 and Figure 1, respectively. The pasting profiles and pasting properties in the table were measured using RVA as described in Section 2.3. Gum Cordia (GC) significantly (*p* < 0.05) decreased the pasting temperature (PT) at 1 and 2% concentrations. The same result was obtained with the addition of gum ziziphus (GZ) at 1% and 2% levels. The effect of the gums on the PT was concentration dependent; a decrease was produced by adding GC at 1%, but the opposite effect was observed at 2% because the PT increased but remained less than the control. Conversely, a higher GZ level further decreased the PT of the blend. This tendency was observed for gums other than the ones tested here, where xanthan gum was reported to decrease the PT even further at greater concentrations, whereas pectin responded similarly to GC, with lower concentrations decreasing the PT less [11]. Although starch is the most important component of wheat flour in terms of pasting characteristics, the presence of other constituents in the flour alters the starch’s pasting properties when compared to pure starch. Consequently, as other researchers have pointed out, some form of interaction between the hydroxyl groups of the starch and the gum is feasible, but it is limited in the flour suspension [34]. The limitation could be due to the interference of the other components of the flour such as proteins, fiber, and lipids. The PT reduction is critical because it signifies an earlier start of starch gelatinization, which translates into more starch available as an enzyme substrate during baking, which has a negative effect on the bread loaf volume.

The swelling of starch granules prior to physical break down is referred to as peak viscosity (PV) [12]. Polymer complexes formed by hydrocolloid interactions with specifically leached amylose molecules, and low-molecular-weight amylopectin molecules have been associated with enhanced viscosity during starch pasting [14,35]. The PV of the flour suspension ranges from 1712 to 1951 cP, as seen in Table 1. When 2% GC was added to flour suspension, the largest gain in PV was from 1886 to 1951 cP, while the maximum loss was from 1886 to 1712 cP when 2% GZ was added. The GZ, once again, had the largest impact on the wheat flour suspension pasting. When the GC was introduced to the wheat starch rather than the flour suspension, the increase in PV was more pronounced, whereas it was less noticeable with the flour suspension.

According to the above findings, the presence of gums caused variations in the maximum PV of the wheat flour paste, which were mostly induced by interactions between the gum and the starch granules. The ability of starch granules to swell freely before physical breakdown is reflected in maximum PV; hence, starch with a high swelling power also has a high maximum PV. According to the data in Table 1, GZ reduced starch granule swelling, whereas 2% GC increased granule swelling, resulting in the maximum PV for the flour suspension. The 1% GC did not appear to promote granule swelling and thus had no effect on PV. Other gums, such as xanthan guar, locust bean gum, and alginate gums, boosted granule swelling and increased the PV [11]. A number of studies have suggested that hydrocolloids have an effect on PV based on two facts: hydrocolloids interact with solubilized starch (amylose) and this action exerts pressure on the granule, causing granules to break down; the second fact is that addition of the thickening agent increases the pressures acting on the granules, increasing granule breakdown and the amount of solubilized starch produced [34,36]. Other researchers consider starch pastes as suspensions of swollen granules dispersed in a continuous macromolecular medium, claiming that the hydrocolloid is only present in the continuous phase of the medium, and that as long as the starch granules swell, the concentration of the hydrocolloid within the continuous phase increases, leading to a significant increase in the continuous phase’s viscosity [37].

The setback was also affected when 1 or 2% GC and GZ gums were added to the wheat flour suspension. In terms of the setback, both GC and GZ caused a significant decrease in this parameter (Table 1), with the 1% GZ having a smaller influence. Amylose chains diffused outside the starch granules retrograde (come together) during the cooling stage of the flour gel. This mechanism is what causes the bread crumb to firm up in the first few hours after baking. As a result, it is practical to include ingredients that induce a reduction in setback, resulting in a delay the firming of the bread crumb. Consequently, GC and GZ added to wheat flour could be deemed efficient antistaling ingredients in the bread-making process. Setback was also claimed to be reduced by locust bean, carrageenan gum, and alginate, while there are conflicting reports on the effect of xanthan gum on setback [38].

The viscosity breakdown shows how much viscosity has been lost. Breakdown values increased significantly (*p* < 0.05) when GC was added but decreased in the presence of GZ. The GC at a concentration of 2% caused the maximum breakdown, while the 2% GZ exhibited the least. The breakdown values changes for both gums were concentration dependent. This result is in agreement with previous reports where some hydrocolloids increased breakdown and other decreased it, which could be attributed to the molecular structure of the gums [39]. The starch granules become less resistant to thermal treatment and mechanical shear as the breakdown values increase. As a result, the addition of Cordia gum improved the flour’s tolerance to heat treatment and mechanical shear. Hydrocolloids with a low Mw and linear structure, such as alginate, have a low breakdown value, whereas blends with branched hydrocolloids have a high breakdown value [38,40].

### 3.2. Textural Properties of Flour Blends

Table 2 presents the textural properties of the examined gels. GC and GZ significantly reduced gel hardness. Higher GC concentrations, in particular, resulted in reduced hardness, but 2% was more effective, whereas 1% GZ was more effective than 2%, although gel hardness remained lower than that in the control. The highest hardness after the control was recorded for samples with 2% GZ and the least with 2% GC. Hardness is induced mostly by starch gel retrogradation, which is influenced by amylose and amylopectin chain rearrangement. As a result, the decrease in hardness as GC concentration increased could be due to GC creating hydrogen bonds with amylose molecules, interfering with the creation of organized structures during starch retrogradation [41]. Adhesiveness is the amount of energy necessary to overcome the attraction forces that exist between the surface of the food and the surfaces of the other materials with which it comes into direct contact [42]. Both gums promoted a decrease in adhesiveness when compared to the control, especially the ones containing 1% GC. The least decrease was recorded for the 2% GZ. The reduction in adhesiveness could be explained by the gums preventing starch–starch interaction [43]. The extent to which a material may be deformed before it ruptures is referred to as cohesiveness. The addition of gums resulted in a significant increase in cohesiveness (*p* < 0.05), ordered as 1% GZ > 2% GC > 2% GZ > 1% GC. This shows that the gel is more resistant to breakdown before it occurs. The energy required for food disintegration before swallowing is directly represented by gumminess, which is the product of hardness x cohesiveness. The gumminess of the gel was significantly increased by GZ, whereas GC produced similar gumminess to the control. This is to be expected, given that GZ had a higher cohesiveness than GC and a similar hardness (Table 2). With the addition of apple fiber, Yildiz, et al. [44] reported that the gumminess, chewiness, and hardness of wheat starch gel decreased. This contradicts this study, given the presence of other components in wheat flour studied here rather than just wheat starch. The gumminess, chewiness, and cohesiveness of the gel were all dominated by the influence of 1% GZ (Table 2), whereas GC had very little effect on the gumminess and chewiness. With the exception of hardness, an increased GC content appeared to have a smaller impact on the gel characteristics.

### 3.3. Dough Mixing Properties

The water absorption represents the amount of water needed for the dough to reach 500 FU. The dough quality of the samples was determined using DoughLab, and the results are shown in Table 3. This study revealed that GC and GZ had different impacts on the wheat flour dough properties. The impact was both gum type and concentration dependent. A limited decrease in the water absorption (WA) was observed for the GC blend, where GZ had no impact on WA. The WA varies depending on the structure of the gum and its ability to absorb water, which interferes with the ability of the flour to absorb water. The reduction in WA caused by GC was identical to that seen with guar gum, as reported in the literature [45]. As a result, it is recommended that GZ be added to maintain the WA of the dough. Dough development time (DDT) refers to the amount of time it takes for flour dough to reach its maximum consistency after it has been mixed. Because the reduction in WA produced by GC was accompanied by a significant reduction in DDT, the addition of GZ resulted in a significant increase in DDT that was not concentration dependent. Both blends had significantly less DDT than the control (Table 3). The DDT was further reduced when the concentration of both gums increased. This study contradicts suggestions that adding hydrocolloids such as xanthan, alginate, or guar increased DDT, although it agrees with the effect of k-carrageenan or HPMC [12,38]. When hydrocolloids are added, it takes longer for the dough matrix to develop, resulting in a higher DDT, and the opposite is true when the DDT is reduced. Dough stability is a measurement of the dough’s capacity to maintain consistency over time and an indicator of its mechanical strength. Both gums significantly reduced dough stability at both concentrations (Table 3); however, the 1% GZ had the smallest decline, 4.03 min vs. 5.70 min for the control. Other gums, such as alginate and xanthan gum, have been reported to reduce the stability of wheat flour dough, where others such as guar gum were reported to increase dough stability [12]. MTI is the difference between the BU at the top of the curve at peak time and the value at the top of the curve 5 min later, which indicates dough softening during mixing. The MTI is a measurement of the flour’s mixing tolerance; an MTI value of 30 B.U. or less is considered extremely good for bread wheat flours. A flour with an MTI greater than 50 FU has a lower mixing tolerance and is more likely to cause problems during mechanical handling and dough preparation. MTI was reduced by the addition of both gums, and increases in gum concentration increased MTI even more, indicating dough softening and reduced mixing tolerance. The negative impact of 1% or 2% GC on MTI was significantly less than that of GZ, but the GC effect was concentration dependent, with the 2% GC having about a 67% stronger effect than the 1%. Other gums such as xanthan, guar, and alginate were reported to decrease the MTI of strong gluten hard red spring wheat flour at 2% and higher, where alginate at 2% reduced MTI to zero [46]. The FQN is the distance in mm from the point of water addition to the point where the height in the center of the curve has dropped 20 BU along the time axis. FQN was significantly (*p* < 0.05) lower for the wheat flour blends with GC and GZ compared to the control, which could be attributed to the gum–gluten interaction. Rice bran and bagasse fiber were reported to lower the FQN of wheat flour due to the gluten–fiber interaction [47,48]. Wheat flour containing 1% GC is still suitable for bread baking according to FQN (Table 3).

### 3.4. Bread and Cake Quality Evaluation

Table 4 shows the quality of the loaves that were enriched with GC and GZ after 24 and 96 h of storage The specific volume of the GC and GZ-supplemented samples was significantly (*p* < 0.05) higher than that of the control, as can be observed. Numerous authors have also reported on the increasing influence of hydrocolloids on bread volume, such as guar gum, cellulose derivative, and locust bean gum [9,12,38,49]. The magnitude of the increase in the loaf specific volume is determined by the type of gum and its concentration; however, GZ increased loaf volume more than GC, while GC increased loaf weight significantly. The capacity of these gums to increase bread volume may be due to the fact that when the hydrated chains of these gums are heated to high temperatures, the water molecules linked with the gum side chains are released, allowing for stronger connections between the chains. As a result, a temporal network is formed, which will collapse during the cooling process. This network will provide support to the dough’s gas cells (during the earliest stages of baking), which will increase during baking, reducing gas losses and increasing the bread volume [50,51].

Bread firmness was significantly lower in the presence of both gums after storage for 24 or 96 h. After 24 h, bread firmness ranked as follows: control > 2% GC > 1% GZ > 2% GZ > 1% GC, and after 96 h: control > 2% GC > 1% GC > 1% GZ > 2% GZ. When compared to the control, there was a clear crumb-softening impact of the gums. Because the inverse relationship between hardness and moisture content has been widely reported, the decreased crumb hardness achieved in the samples containing gums could be due to the high moisture content of these loaves. This characteristic cannot be related solely on the water absorption during dough formation because the control, 1%, and 2% GZ exhibited the highest water absorption during dough mixing, but after 24 h, 1% GC exhibited the least firm bread crumbs. However, after 96 h, the 1% GC and 2% GZ had the least firm bread crumbs. As a result, firmness is entirely dependent on the dough’s capacity to preserve the moisture content throughout the baking process, which may explain why the 2% GZ had the least firm crumbs after 96 h of storage. These data are in agreement with published data on the effect of a number of hydrocolloids on bread firmness [52]. Furthermore, the GC or GZ side chains may interfere with interactions between the starch–starch polymers (prevention of retrogradation) and proteins–starch, resulting in softer crumbs [53]. As evidenced by the control, there was a noticeable increase in hardness with or without the gum. However, because the hardness of the gum-containing crumbs was always lower than that of the control, the gums produced softer crumbs over time compared to the control. The springiness of the bread crumbs was immeasurable. The images in Figure 2 illustrate the bread crumb grain density of the control bread crumb as it was modified by the added gum. When compared to the control, gum-containing samples seemed to be less dense. Furthermore, the gum content had an effect on the crumb image, as seen by the difference in the 1 and 2% gum content. On the other hand, the pores in the composite bread increased as a result of the inclusion of 2% GC and 1% GZ. These findings are consistent with the greater specific volume recorded in Table 4 for these loaves.

Cake volume is one of the most important quality attribute as it influences consumer acceptance of the final product. Moreover, it is directly affected by the ingredients used, especially those with a direct influence on batter aeration and foam stability, such as the fat content. The control had the highest volume, followed by the sample containing 2% GC, whereas the sample containing 1% GC had the lowest volume (Table 4). The increased viscosity of the batter, which slows air diffusion and permits it to be held early in the baking process, could account for the positive influence of the GZ gum on the cake volume. Because starch gelatinization temperature and batter viscosity are interrelated, increasing the gelatinization temperature improves batter viscosity and consequently cake volume [11]. By elevating the starch gelatinization temperature, xanthan and guar gums, for example, were shown to increase batter viscosity, but pectin and cellulose derivatives (HPMC) decreased it [54]. We previously reported that both GC and GZ gums increased the gelatinization temperature of wheat starch [29], implying that an increase in cake volume would be predicted in the presence of GC and GZ gums; however, the data did support this hypothesis with respect to loaf volume but not the specific volume. The fact that the gelatinization temperature of wheat starch had no influence on cake volume is remarkable because neither gum created a higher specific cake volume than the control. Yet, by raising the gelatinization temperature of wheat starch, the 2% GC came near the specific volume of the control (Table 4). The texture analysis results for one-and four-day-old cake samples are shown in Table 4. All textural qualities changed significantly during storage. The inclusion of gums at 1% GC significantly increased the crumb firmness of the cake after 24 h (*p* > 0.05), whereas 2% GC or 2% GZ reduced it, and 1% GZ had no effect. Unlike after 24 h, the 1% GZ reduced cake firmness significantly after 96 h, although 1% GC and 2% GZ gums were unsuccessful in lowering firmness after 24 h. As a result, the 2% GC generated the least firm cake after 24 and 96 h, while the control produced the most firm cake after both storage times. Gum-containing samples had significantly increased springiness regardless of storage period. The highest springiness was found under 2% GC, followed by 1% GZ, while the least springiness was observed in the control. Cake firmness was reported to be inhibited by alginate and locust bean gums but increased by xanthan and guar gums, indicating that cake firmness is dependent on the hydrocolloid type [55].

The varying impacts of hydrocolloids on cake firmness must be explained by the various chemical interactions between hydrocolloids and starch that alter starch retrogradation [56]. It is remarkable that there is no evident relationship between the water content of the cakes and their firmness. Figure 3 illustrates the cake images of the control and composites. The GZ composite cake appeared to be shorter in height, but it had the highest volume and weight, as shown in Table 4.

### 3.5. Crumb Color of Bread and Cakes

The presence of gums changed the color of the crust of the bread or cake, as shown by the change in L* values in Table 5. Significant changes (*p* < 0.05) in crust color were detected in bread and cake containing GC and GZ gums. The control bread and cake had the lightest crust color (high L* value), whereas the bread or cake produced with gums had the darkest crust color (lowest L*) (Table 5). The color of the GC bread or cake samples appeared to be darker (lower L*), whereas the color of the GZ samples seemed to be lighter (higher L*). Meanwhile, the L*-value of the control bread crust external surface decreased linearly from 79 to 72 for the bread and from 81 to 73 for the cake, indicating increased surface browning due to gums, while a* variations revealed a continuous increase and stabilization of the red coloration, and b* reached a maximum of yellow coloration for the control and then dropped due to the addition of gums. Even with varying baking settings, these results are within the same range as those published in the literature, where the L* for French bread was found to be around 75 [57].

### 3.6. Sensory Evolution of Bread and Cake

The mean findings of the hedonic sensory evaluation of bread made with the GC and GZ gums examined are shown in Figure 4. In terms of general acceptability and crumb color, it can be stated that the inclusion of gums, with the exception of 2% GC, resulted in panelists appreciating bread-composites more than the control. When GZ was used, the greatest overall sensory score was attained. The control bread received the lowest scores in the texture hedonic evaluation, while the crust color of the control bread received the highest score. As a result, the panelists thought that GZ was a better choice than GC and in certain circumstances even better than the control. The sensory score for the cake is shown in Figure 5. Cake overall acceptability, texture, porosity, crumb color aroma, and taste were all lowest for 2% GC, whereas overall acceptability, texture, porosity, and crumb color were highest for the control and 1% GZ. As with the bread, the 1% GC performed better than the 2% GC in relation to the sensory evaluation.

## 4. Conclusions

The dough water absorption and dough development time were reduced by gum Cordia, whereas the dough stability and mixing tolerance index (MTI) were reduced by both GC and GZ gums, while increasing gum concentration increased MTI even more, indicating dough softening. As can be seen, the specific volume of the GC and GZ-supplemented samples was considerably higher (*p* < 0.05) than that of the control. Gum ziziphus, on the other hand, raised the loaf volume more than gum Cordia. After 24 or 96 h, bread firmness was significantly decreased in the presence of both gums. The control cake had the largest volume, followed by the sample with 2% GC; however, the sample with 1% GC had the lowest volume. During storage, all of the textural features of the cake changed significantly. The addition of gums at 2% GC considerably increased the cake’s crumb firmness after 24 h, whereas 2% GC or GZ decreased it. The control bread and cake had the darkest crust (lowest L* value), whereas the addition of gums resulted in a lighter crust. The influence of hydrocolloids on the sensory quality of cake and bread was highly dependent on the type of gum used. However, it can be inferred that adding gums to the bread and, to a lesser extent to the cake, increased the overall sensory score, especially when 1% GZ or GC was applied. Only the 2% gum Cordia received a significantly worse overall approval score than the control bread or cake. As a result, 2% GZ is preferred for bread and 2% GC for cake in terms of firmness, whereas 2% GZ is recommended for both bread and cake in terms of overall acceptability, texture, and crumb color. Furthermore, since a distinct difference between 1% and 2% gums had a substantial influence on baked goods, higher quantities of gum may generate even better results.

## Figures and Tables

**Figure 1 foods-11-00460-f001:**
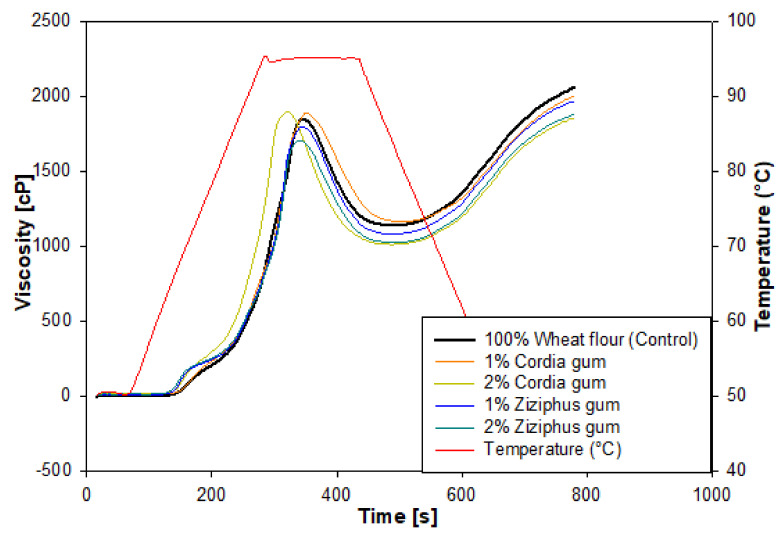
RVA profiles of wheat flour blends with Cordia and Ziziphus gums.

**Figure 2 foods-11-00460-f002:**
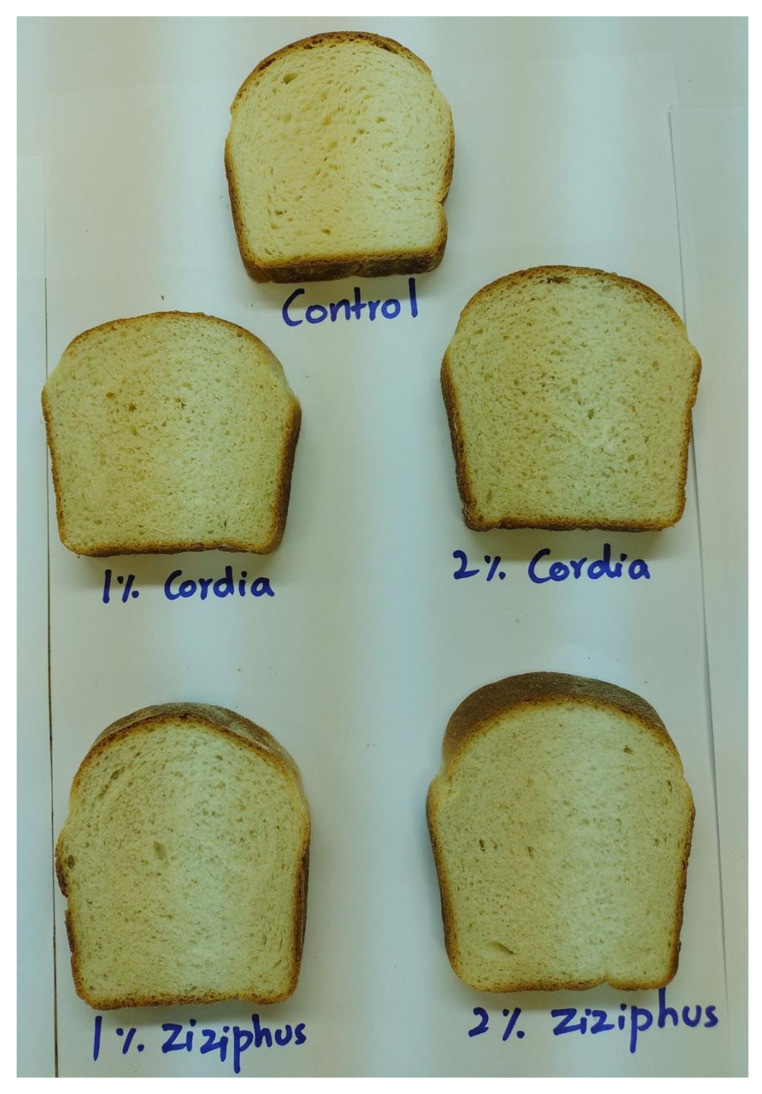
Images of the central slices of the different breads containing Cordia and Ziziphus gums.

**Figure 3 foods-11-00460-f003:**
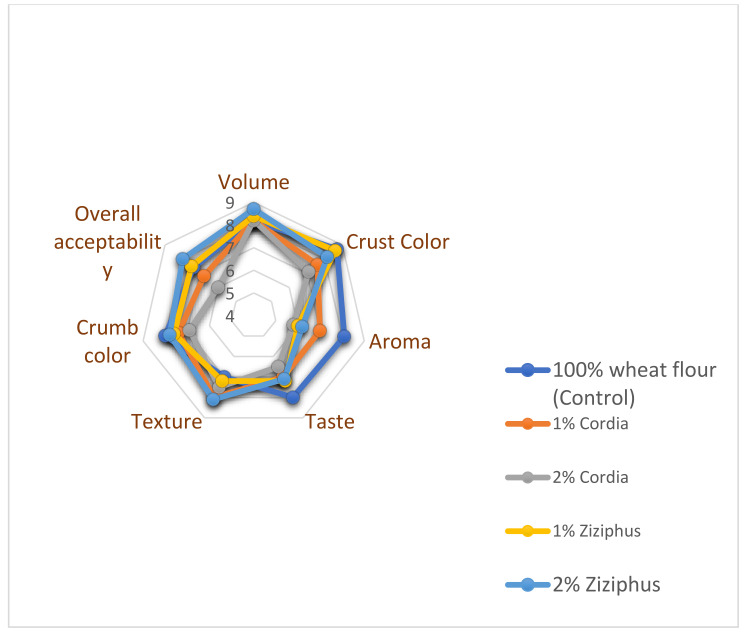
Sensory score based on the 9-point hedonic scale of bread containing Cordia and Ziziphus gums.

**Figure 4 foods-11-00460-f004:**
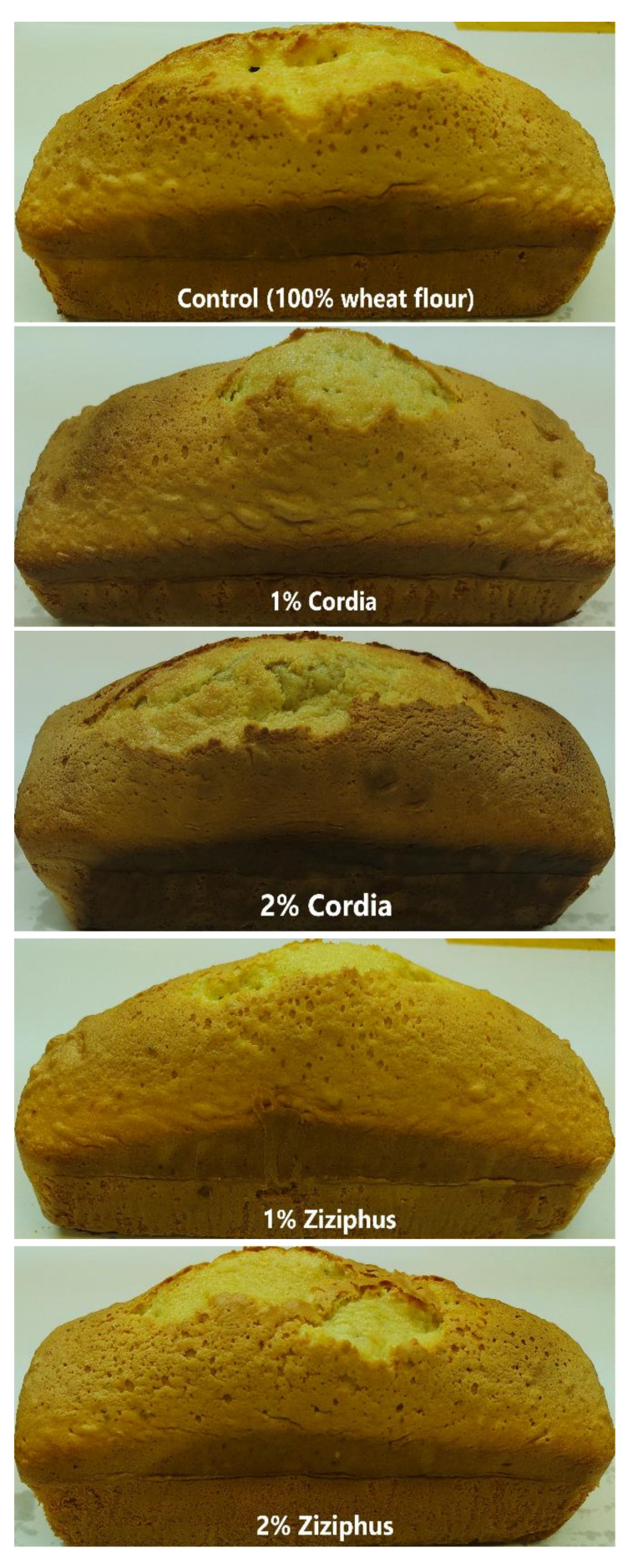
Images of the cake loaves containing Cordia and Ziziphus gums.

**Figure 5 foods-11-00460-f005:**
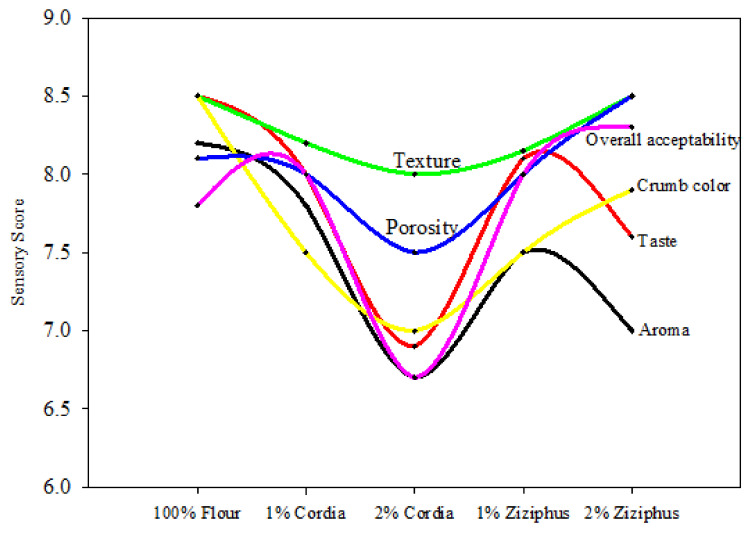
Sensory score of cake containing Cordia and Ziziphus gums.

**Table 1 foods-11-00460-t001:** Effect of Cordia and Ziziphus gums on the pasting properties of wheat flours.

	^2^PV (cP)	^3^BD (cP)	^4^FV (cP)	^5^SB ^6^(cP)	^7^PT (°C)
Control (100% WF^1^)	1886 ± 35.26 ^b^	736 ± 21.39 ^bc^	2056 ± 1.70 ^a^	905 ± 12.36 ^a^	69.40 ± 0.04 ^a^
1% Cordia gum	1885 ± 2.87 ^b^	763 ± 32.32 ^b^	1968 ± 21.67 ^b^	813 ± 49.71 ^c^	66.93 ± 0.61 ^b^
2% Cordia gum	1951 ± 41.67 ^a^	918 ± 23.72 ^a^	1869 ± 12.28 ^c^	834 ± 5.89 ^bc^	67.37 ± 0.33 ^b^
1% Ziziphus gum	1776 ± 18.80 ^c^	709 ± 8.65 ^cd^	1950 ± 11.84 ^b^	883 ± 1.70 ^ab^	66.13 ± 0.02 ^c^
2% Ziziphus gum	1712 ± 7.93 ^d^	674 ± 3.68 ^d^	1893 ± 14.82 ^c^	849 ± 2.62 ^bc^	65.65 ± 0.35 ^c^

1 WF = wheat flour; 2 PV = Peak Viscosity; 3 BD = Breakdown viscosity; 4 FV = Final Viscosity; 5 Sb = Setback viscosity; 6 cP = Centipoise; 7 PT = Pasting Temperature; Values followed by different letters in columns are significantly different at p ˂ 0.05.

**Table 2 foods-11-00460-t002:** Effect of Cordia and Ziziphus gums on the properties of wheat flour gel.

	Hardness (N)	Gumminess (N)	Chewiness (N.mm)	Springiness (mm)	Cohesiveness	Adhesiveness (mJ)
Control ^1^	0.60 ± 0.01 ^a^	0.29 ± 0.01 ^b^	2.93 ± 0.01 ^b^	10.00 ± 0.08 ^a^	0.49 ± 0.01 ^d^	0.83 ± 0.05 ^a^
1% Cordia	0.56 ± 0.01 ^bc^	0.29 ± 0.01 ^b^	2.90 ± 0.06 ^b^	9.85 ± 0.09 ^b^	0.52 ± 0.01 ^c^	0.27 ± 0.04 ^c^
2% Cordia	0.52 ± 0.00 ^d^	0.29 ± 0.01 ^b^	2.81 ± 0.01 ^c^	9.73 ± 0.05 ^c^	0.55 ± 0.01 ^b^	0.37 ± 0.05 ^c^
1% Ziziphus	0.54 ± 0.01 ^c^	0.32 ± 0.00 ^a^	3.12 ± 0.04 ^a^	9.87 ± 0.04 ^b^	0.58 ± 0.00 ^a^	0.33 ± 0.09 ^c^
2% Ziziphus	0.57 ± 0.01 ^b^	0.30 ± 0.01 ^b^	2.90 ± 0.04 ^b^	9.73 ± 0.05 ^c^	0.53 ± 0.01 ^c^	0.67 ± 0.05 ^b^

^1^ 100% Wheat flour; Values followed by different letters in columns are significantly different at *p* ˂ 0.05.

**Table 3 foods-11-00460-t003:** Effect of Cordia and Ziziphus gums on the dough mixing properties.

	^2^WA (%)	^3^DDT (min)	Stability (min)	Softening ^4^(FU)	^5^MTI (FU)	^6^FQN (mm)
Control^1^	55.27 ± 0.09^a^	1.60 ± 0.08^b^	5.70 ± 0.22^a^	91.67 ± 2.36^d^	35.67 ± 4.19^c^	61.23 ± 0.95^a^
1% Cordia	54.33 ± 0.12^b^	1.40 ± 0.16^c^	3.53 ± 0.78^b^	75.00 ± 4.08^e^	42.00 ± 5.89^c^	59.47 ± 1.08^a^
2% Cordia	54.03 ± 0.09^c^	1.13 ± 0.21^c^	2.20 ± 0.22^c^	100 ± 4.08^c^	70 ± 0.00^b^	52.30 ± 0.78^b^
1% Ziziphus	55.25 ± 0.04^a^	3.53 ± 0.12^a^	4.03 ± 0.29^b^	112.30 ± 2.58^b^	85.33 ± 4.11^a^	48.07 ± 041^c^
2% Ziziphus	55.20 ± 0.08^a^	3.43 ± 0.05^a^	3.50 ± 0.08^b^	126.30 ± 2.58^a^	89.00 ± 1.41^a^	46.20 ± 0.57^d^

^1^ 100% wheat flour; ^2^ WA = Water absorption; ^3^ DDT = Dough development time; ^4^ FU = Farino units; ^5^ MTI = mixing tolerance index; ^6^Farinograph Quality Number; Values followed by different letters in columns are significantly different at *p* ˂ 0.05.

**Table 4 foods-11-00460-t004:** Effect of Cordia and Ziziphus gums on the volume, weight, and firmness of bread and cake samples.

	Loaf Volume (cm^3^)	Loaf Weight (g)	Specific Volume (cm^3^/g)	Firmness (g) 24 Hours	Springiness (%) 24 Hours	Firmness (g) 96 Hours	Springiness (%) 96 Hours
	Pan bread						
Control (100% WF)	791.67 ± 8.50 ^e^	310.00 ± 0.82 ^c^	2.55 ± 0.03 ^e^	865.84 ± 23.81 ^a^	^1^-	2333.05 ± 92.65 ^a^	^1^-
1% Cordia gum	900.00 ± 8.16 ^d^	325.17 ± 1.03 ^a^	2.77 ± 0.03 ^d^	251.35 ± 9.89 ^d^	-	645.41 ± 32.05 ^bc^	-
2% Cordia gum	968.33 ± 6.24 ^b^	323.00 ± 0.14 ^a^	2.99 ± 0.02 ^c^	361.55 ± 16.22 ^b^	-	762.54 ± 65.44 ^b^	-
1% Ziziphus gum	953.33 ± 4.71 ^c^	311.33 ± 0.47 ^c^	3.07 ± 0.01 ^b^	301.07 ± 8.69 ^c^	-	623.99 ± 28.33 ^c^	-
2% Ziziphus gum	985.00 ± 4.08 ^a^	313.67 ± 0.47 ^b^	3.14 ± 0.01 ^a^	238.86 ± 10.93 ^d^	-	519.99 ± 27.49 ^c^	-
	Sponge cake						
Control (100% WF)	740 ± 10.21 ^c^	261 ± 0.89 ^d^	2.84 ± 0.02 ^a^	221.60 ± 10.48 ^b^	52.54 ± 0.04 ^c^	402.81 ± 12.80 ^a^	49.65 ± 0.23 ^c^
1% Cordia gum	718 ± 5.36 ^d^	272 ± 1.23 ^c^	2.64 ± 0.01 ^e^	265.13 ± 07.48 ^a^	53.69 ± 0.23 ^b^	384.79 ± 06.07 ^ab^	50.91 ± 0.08 ^b^
2% Cordia gum	711 ± 7.36 ^d^	258 ± 0.80 ^d^	2.75 ± 0.05 ^b^	191.07 ± 13.92 ^c^	55.44 ± 0.44 ^a^	262.02 ± 16.79 ^c^	51.65 ± 0.26 ^a^
1% Ziziphus gum	820 ± 4.20 ^a^	300 ± 1.23 ^a^	2.73 ± 0.02 ^c^	225.88 ± 08.12 ^b^	54.53 ± 0.96 ^a^	361.90 ± 18.53 ^b^	51.36 ± 0.79 ^a^
2% Ziziphus gum	784 ± 6.32 ^b^	288 ± 2.10 ^b^	2.71 ± 0.03 ^d^	203.45 ± 09.20 ^c^	52.71 ± 0.60 ^c^	385.53 ± 19.00 ^ab^	48.58 ± 0.99 ^d^

^1^ No springiness reading was recorded for the bread; Values followed by different letters in columns (under bread or cake) are significantly different at *p* ˂ 0.05.

**Table 5 foods-11-00460-t005:** Effect of Cordia and Ziziphus gums on the crumb color parameters of bread and cake samples.

	^2^L*	^3^a*	^4^b*
	Pan bread		
^1^Control	79.43 ± 0.39 ^a^	−7.39 ± 0.06 ^d^	21.11 ± 0.16 ^a^
1% Cordia gum	73.14 ± 1.02 ^cd^	−6.68 ± 0.09 ^c^	19.54 ± 0.36 ^b^
2% Cordia gum	72.79 ± 0.37 ^d^	−6.31 ± 0.11^b^	19.94 ± 0.39 ^b^
1% Ziziphus gum	74.59 ± 0.98 ^c^	−6.79 ± 0.05 ^c^	18.50 ± 0.08 ^c^
2% Ziziphus gum	76.41 ± 0.81 ^b^	−6.08 ± 0.02 ^a^	18.25 ± 0.10 ^c^
	Sponge cake		
Control^1^	81.53 ± 0.20 ^a^	−9.16 ± 0.07 ^e^	26.79 ± 0.12 ^a^
1% Cordia gum	75.78 ± 0.61^d^	−7.34 ± 0.13 ^b^	24.69 ± 0.14 ^b^
2% Cordia gum	72.72 ± 0.52 ^e^	−6.12 ± 0.08 ^a^	23.59 ± 0.12 ^c^
1% Ziziphus gum	79.65 ± 0.27 ^c^	−8.49 ± 0.21^d^	24.82 ± 0.05 ^b^
2% Ziziphus gum	80.44 ± 0.38 ^b^	−8.18 ± 0.12 ^c^	23.56 ± 0.18 ^c^

^1^ 100% wheat flour; ^2^ L* = lightness; ^3^ a* = green/red; ^4^ b* = blue/yellow; Values followed by different letters in columns (under bread or cake) are significantly different at *p* ˂ 0.05.

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
