# Peer review of "Functionality of Cordia and Ziziphus Gums with Respect to the Dough Properties and Baking Performance of Stored Pan Bread and Sponge Cakes"

_foods, 2022, doi:10.3390/foods11030460_

Round 1

Reviewer 1 Report

The authors undertook to investigate the effect of the addition of Cordia gum and Ziziphus gum on the quality of two baked goods. The authors have very well chosen methodology to show the differences between the variants of the samples obtained. The discussion of the results presents a good level and the conclusions follow from the research presented in the manuscript.

However, the paper has many imperfections that need to be corrected or filled with information. First of all, the introduction of the paper needs to be reorganized and completed (according to the comments below). The information in the methodology should be supplemented by giving an exact number of sample variants and the number of repetitions of the analysis performed as well as a description of the statistical analysis performed on the obtained data. It is also necessary to work on graphs and tables and their proper placement in the text. Authors should update the list of references with recent publications. Conclusions are drawn from the research, but there is no clear answer as to which gum and in what proportion in the product is best in practice; there are also no recommendations about future utilization of studied gums.

Abstract

line 22- sentence need verification, because authors are contradicting themselves

line 23- use abbreviation of gums or full name through all abstract

line 24- sentence need correction (what does it mean “…within reason…”)

Introduction

all introduction should be rewritten – authors must give the latest advances in the topic of study, show the scientific problem that they want to resolve. Now is chaotic, gathered data about different hydrocolliods. Authors don’t mention about characteristic properties of Cordia and Ziziphus gums and why they want use this gums in their study.

line 34 – please correct the form how you refers to someone results (see the authors guide)

line 82- please make here another paragraph to extract aim of the study

line 86- I recommend to reformulate title of manuscript in order to contain information that storage tests were performed

Materials and Methods

Isolation of gums – authors should add brief description of procedure

In materials and methods authors should justify the usage of 1 and 2 % level of gums addition to studied products. Why 1% and not 0,5% or 3%?

line 93-94 – authors should mention here all tests that were performed on flour-gum mixtures because the numbering of sections don’t show what was done on flour and what on bread and cake

line 96-101 – authors don’t mention what they measure here to define pasting properties of samples

line 102 - gel texture – authors must give the specification of the probe used for TPA analysis and the model of device

line 117- add that method is AACC method

line 119 – specify what is “improver”, are these gumes? please verify; add information about bread variants made in this study  

line 124- please provide oven model and manufacturer

line 127- add information about adding gums and in what percentage they were added? give information how many cake variants was made in this study

line 129- please provide oven model and manufacturer

line 148- please provide information how the evaluators were trained for sensory evaluation of food, or maybe only to bread and cake only? Were they professional evaluators?

There must be added section about used statistical methods and software to develop obtained results.

Also everywhere in the methods description must be added number of repetitions.

Results and Discussion

The authors cite old literature, so they need to supplement the discussion of results with publications from the last 5 years. Here I provide some suggested publications that can be used in discussion in almost all section of this manuscript:

https://doi.org/10.1111/jfpp.16104

https://doi.org/10.3390/molecules26154641

https://doi.org/10.1016/j.lwt.2021.112156

https://doi.org/10.1177/1082013220980594

Table 1 is too far away situated from first citing and discussion based on that table. Please rearrange the position of the table in the text.

Authors don’t mention if the properties in table 1 were measured by device or were calculated on base of certain parameters.

line 163, 209 - authors cite manuscript that is under review -papers not accepted for publication can not be cited ! please delete this reference

lines 209-213 – this sentence needs more references

line 300- MTI abbreviation as other abbreviations used for the first time in the text should be expanded

Authors should made separate section about results from storage tests of bread and cake.

Figure 3 needs formatting and deletion of repetitive title.

figure 4 first picture – change the description on ‘Control (100% wheat flour)’

line 443 – please change word magenta on red

Conclusion

line 522-523- I recommend to delete this sentence because it is repetition of informations already given in manuscript.

In this section there is lack of recommendation about future utilization of gums in bread and cake making including % content of gums. There is no definite answer given as to whether the addition of gums improved the quality of baked goods and which gum is better to use.

Author Response

line 34 – please correct the form how you refers to someone results (see the authors guide) Done

line 82- please make here another paragraph to extract aim of the study. Done

line 86- I recommend to reformulate title of manuscript in order to contain information that storage tests were performed. Done

Materials and Methods

Isolation of gums – authors should add brief description of procedure. Done

In materials and methods authors should justify the usage of 1 and 2 % level of gums addition to studied products. Why 1% and not 0,5% or 3%? Explanation was given

line 93-94 – authors should mention here all tests that were performed on flour-gum mixtures because the numbering of sections don’t show what was done on flour and what on bread and cake. It was clarified

line 96-101 – authors don’t mention what they measure here to define pasting properties of samples. Measured parameters were given

line 102 - gel texture – authors must give the specification of the probe used for TPA analysis and the model of device. Given

line 117- add that method is AACC method. Done

line 119 – specify what is “improver”, are these gumes? please verify; add information about bread variants made in this study.   Done

line 124- please provide oven model and manufacturer. Done

line 127- add information about adding gums and in what percentage they were added? give information how many cake variants was made in this study. Explained

line 129- please provide oven model and manufacturer. Done

line 148- please provide information how the evaluators were trained for sensory evaluation of food, or maybe only to bread and cake only? Were they professional evaluators? Explanation was given

There must be added section about used statistical methods and software to develop obtained results. Section was added

Also everywhere in the methods description must be added number of repetitions. Done

Results and Discussion

The authors cite old literature, so they need to supplement the discussion of results with publications from the last 5 years. Here I provide some suggested publications that can be used in discussion in almost all section of this manuscript: Recommended literature was added

https://doi.org/10.1111/jfpp.16104

https://doi.org/10.3390/molecules26154641

https://doi.org/10.1016/j.lwt.2021.112156

https://doi.org/10.1177/1082013220980594

Table 1 is too far away situated from first citing and discussion based on that table. Please rearrange the position of the table in the text. Done

Authors don’t mention if the properties in table 1 were measured by device or were calculated on base of certain parameters. Done

line 163, 209 - authors cite manuscript that is under review -papers not accepted for publication can not be cited ! please delete this reference Done

lines 209-213 – this sentence needs more references. Done

line 300- MTI abbreviation as other abbreviations used for the first time in the text should be expanded. Done

Authors should made separate section about results from storage tests of bread and cake. Done

Figure 3 needs formatting and deletion of repetitive title. Done

figure 4 first picture – change the description on ‘Control (100% wheat flour)’ Done

line 443 – please change word magenta on red. Done

Conclusion

line 522-523- I recommend to delete this sentence because it is repetition of information already given in manuscript. Done

Reviewer 2 Report

In the case of the positive opinion of other reviewers, I list some necessary comments:

If the results are based on statistical significance please refer to the applied method?

Improving English is a priority, there are also too many technical and grammatical errors.

For example:

Line 110:  4±0.01 g

The abbreviation of all mixing properties must be included in 2.5.

The uniformity of presented results in figures must be improved.

Presenting the same results in Tables and Figures is redundant.

Author Response

Improving English is a priority, there are also too many technical and grammatical errors.

For example:

Line 110:  4±0.01 g . Done

The abbreviation of all mixing properties must be included in 2.5. Was done

The uniformity of presented results in figures must be improved. Done

Presenting the same results in Tables and Figures is redundant. OK it was done

Round 2

Reviewer 1 Report

The manuscript was improved but still some issues need to be corrected. I give details below. In some places the english need to be improved.

line 111- please make here another paragraph to extract aim of the study

line 129 – please add citations of these researches

line 161-170- please add bread variants with GC and GZ gums like you did for cake at line 172-173

reference nr 29 - authors cite manuscript that is under review -papers not accepted for publication cannot be cited ! please delete this reference

Figure 3 needs formatting and deletion of repetitive title

figure 4 first picture – change the description on ‘Control (100% wheat flour)

line 562-563- I recommend to delete this sentence because it is repetition of informations already given in manuscript.

Author Response

The manuscript was improved but still some issues need to be corrected. I give details below. In some places the English need to be improved.

Thank you for taking the time 

line 111- please make here another paragraph to extract aim of the study.

Paragraph was added

line 129 – please add citations of these researches

The references are there in each testing method section; 2.3, 2.4, 2.5, 2.8, 2.10

line 161-170- please add bread variants with GC and GZ gums like you did for cake at line 172-173

Done

reference nr 29 - authors cite manuscript that is under review -papers not accepted for publication cannot be cited ! please delete this reference

full citation was added to the reference

Figure 3 needs formatting and deletion of repetitive title

Repetition was removed

figure 4 first picture – change the description on ‘Control (100% wheat flour)

Done

line 562-563- I recommend to delete this sentence because it is repetition of informations already given in manuscript.

Was done

Reviewer 2 Report

Line 154: Again, please correct a mass sample…

In figure 1, please correct units….[cP] instead „centipoise“, [s] instead „seconds“

Author Response

Comments and Suggestions for Authors

Thank you for taking the time

Line 154: Again, please correct a mass sample…

Done

In figure 1, please correct units….[cP] instead „centipoise“, [s] instead „seconds“

Done